# A Simple Approach to the Noisy Label Problem Through the Gambler's Loss

## Abstract

Learning in the presence of label noise is a challenging yet important task, and it is crucial to design models that are robust when part of the dataset are mislabeled. In this paper, we discover that a new class of loss functions called the gambler's loss provides strong robustness to label noise across various levels of corruption. Training with this modified loss function reduces memorization of data points and is a simple yet effective method to improve robustness and generalization. Moreover, using this loss function allows us to derive an analytical early stopping criterion that accurately estimates when over-memorization begins to occur. Our overall approach achieves strong results and outperforming existing baselines.

## 1 Learning from Noisy Labels

As a first step of supervised learning tasks, the user often collects a large amount of data in the form of $(x, y)$ pairs where $x$ represents the input data and $y$ the desired labels. However, parts of real world data can often be mislabeled due to 1) annotator mistakes as a natural consequence of large-scale crowdsourcing procedures (Howe, 2008), 2) the difficultly in fine-grained labeling across a wide range of possible labels (Russakovsky et al., 2015), 3) subjective differences when annotating emotional content (Busso et al., 2008), and 4) the use of large-scale weak supervision (Dehghani et al., 2017). Learning in the presence of noisy labels using neural networks is challenging since overparametrized neural networks are known to be able to memorize all labels even with strong regularization (Zhang et al., 2017). When the model memorizes the noisy labels, its generalization deteriorates (e.g., see Figure 1 and Section 1.1). In fact, this agrees with the theoretical finding in (Nakkiran et al., 2019) that neural networks trained with SGD often learns functions stage by stage from simpler functions to more complex ones, and functions whose output is uniformly random are the functions with highest complexity, and so memorization of random labels often come late in the training. Based on these, we may devise special methods to extract signal from noisy labels. Similar to Patrini et al. (2017); Yu et al. (2019); Han et al. (2018), we deal with an idealized noisy label setting, where a noise transition matrix is invoked to define the problem, and each label has the probability defined by the transition matrix to be flipped to another label (see appendix for a formal mathematical definition).

**Organization:** In this section, we first hypothesize and then empirically verify the existence of three stages during training in the presence of noisy labels; in Section 2, we propose that the gambler's loss function is a *noise robust loss function* that can be used for noisy label problems, and we further propose an *analytical early stopping criterion* using the gambler's loss. We then verify our methods through a series of experiments and comparisons with existing approaches.

### 1.1 Training Trajectory of Noisy Labels

In practice, one often faces the problem of learning from a dataset $\mathcal{D} = \{x, y\}_{i=1}^{n}$ whose labels are partially corrupted, i.e. $\mathcal{D} = \mathcal{D}_{clean} \cup \mathcal{D}_{\text{corrupt}}$. We consider a setting where the labels are symmetrically corrupted (i.e. any label has equal probability to be flipped to any other class). Let $\epsilon = 1 - r$ denote the probability that a data point $(x, y)$ is clean, $r$ being the probability of mislabeling. A classification problem can be denoted as learning a function $f(\cdot) : X \to Y$, where $x \in X = \mathbb{R}^n$ are the data points, and $y \in Y = \mathbb{R}^m$ are the targets. In practice, we parametrize $f$ with parameters $\theta$ (i.e. $f_\theta$) and learn $\theta$ by minimizing the empirical risk $\mathbb{E}\left[\ell(f_\theta(x), y)\right]$ for some loss function $\ell$. The standard loss function for classification is cross entropy (or the *nll* loss). However, naive training on $\mathcal{D}$ using *nll* loss often results in a "peaked" learning curve as shown in Figure 1b. We observe that such learning curves often consist of three stages (See Figure 1):

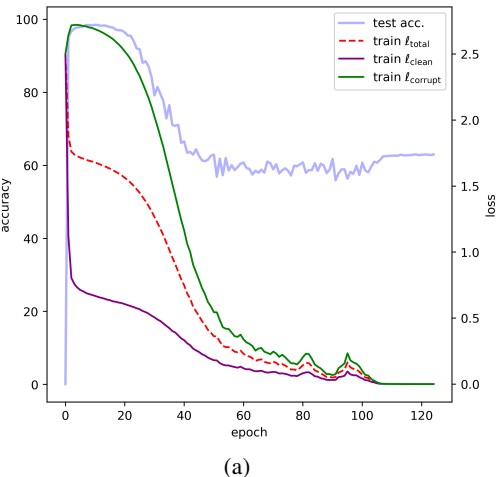 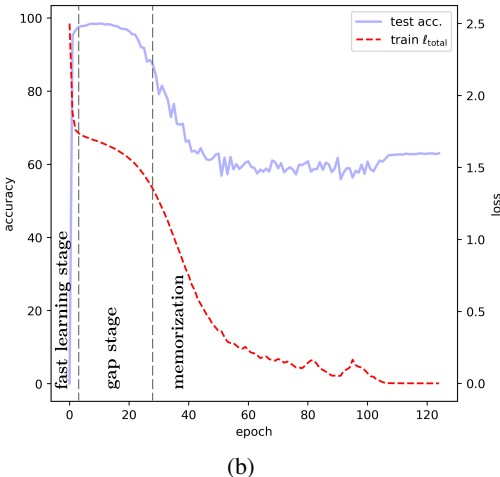

(a)  (b)

Figure 1: Different stages in the presence of label noise. (a) We plot $\ell_{\text{total}}$ (total loss), $\ell_{\text{clean}}$ (loss on $\mathcal{D}_{\text{clean}}$), and $\ell_{\text{corrupt}}$ (loss on $\mathcal{D}_{corrupt}$), we see that during the gap stage, there is a clear "gap" between the $\ell_{\text{clean}}$ and $\ell_{\text{corrupt}}$ where training on clean labels has completed but training on noisy labels has barely started; (b) Hypothesized qualitative division of the three stages: fast learning stage, gap stage, and the memorization stage. Experiment done on MNIST with corruption rate 0.5.

1) **Fast Learning Stage**: The model *quickly* learns the underlying mapping from data to clean labels; one observes rapid decreases in training loss and increases in test accuracy.

2) **Gap Stage**: From Figure 1a, we notice learning on the clean labels is almost complete ($\ell_{clean} \sim$ 0.5) but training on noisy labels has not started yet ($\ell_{\text{corrupt}} \sim 2.5$), and a large gap in training loss exists between $\mathcal{D}_{clean}$ and $\mathcal{D}_{\text{corrupt}}$. Both the train loss and the test accuracy reach a plateau, and this is the time at which the generalization performance is the best.

3) **Memorization**: When there are only noisy labels left to learn from, the model will *memorize* these noisy labels and the train loss decreases slowly to 0. However, memorizing noisy labels *hurts generalization* and the test accuracy decreases as well.

These stages seem to be present across datasets and model architectures when label noise is present. This is problematic because the testing accuracy reaches a peak early in training and then decreases steadily because the noisy gradient from $\mathcal{D}_{\text{corrupt}}$ starts to dominate that from $\mathcal{D}_{clean}$, leading to memorization of noisy labels and affecting generalization. Therefore, tackling the problem of learning from noisy labels involves reducing the negative effect from $\mathcal{D}_{\text{corrupt}}$. This is hard in general, since one does not know a priori which points are mislabeled. We mainly deal with symmetric noise, and we denote the corruption rate (the probability that a data point has a label that is not its true label) as $r = 1 - \epsilon$, where $\epsilon$ is the clean rate. Various approaches have been proposed (Patrini et al., 2017; Han et al., 2018; Yu et al., 2019), but, unfortunately, the best current methods can do is only to slow down the drop in test accuracy instead of preventing it completely (Han et al., 2018).

Our method will be based on the above observation; to summarize the key point, we note:

**Observation.** *When training on a label-corrupted dataset, there is a stage when the average loss on the correct points is much lower that of the corrupted points.*

Intuitively, since features are easier to learn than random noises on average. This phenomenon also motivates a label noise method sometimes called the "small loss method" (Han et al., 2018; Yu et al., 2019), in which the data points with loss higher than a threshold is screened to prevent the network from learning wrong information from such point. The observation also holds theoretically. As shown in (Nakkiran et al., 2019), neural networks trained with SGD often learns functions stage by stage from simpler functions to more complex ones, and functions whose output is uniformly random are the functions with highest complexity. From this observation, we make the following idealized assumption that our method is based on:

**Assumption.** *(Idealized gap stage) During the gap stage, the model has not learned anything about the data points in $\mathcal{D}_{\text{corrupt}}$, and predicts strictly uniform score across all the labels, and it achieves perfect accuracy on $\mathcal{D}_{clean}$.*

We find that this assumption holds well for simple datasets such as MNIST and on datasets with very high corruption rate, where our method achieves best results, and less so on more complicated

datasets such as CIFAR10 (for example, on gap stage, the training accuracy on the clean dataset do not reach close to $100\%$).

## 2 THE GAMBLER'S LOSS FUNCTION AND HOW TO EARLY STOP

We propose an approach that uses the gambler's loss function (Ziyin et al., 2019). Intuitively speaking, the gambler's loss builds on the analogy that making a classification given a data point is the same as a gambler making a bet on a horse race given the horses, and that minimizing the nll loss in the classification problem is the same as maximizing the doubling rate of wealth. The gambler's loss may be generalized to include a reservation option in which the gambler can play "safe" and reserve part of his money in pocket. We refer to this generalized version as the *gambler's loss*. Mathematically, the gambler's loss augments the target space by an "reservation" dimension: $Y \to Y' = \mathbb{R}^{m+1}$, where the $(m+1)$-th dimension is defined as the reservation score. In the case of a neural network, using this loss involves adding one more output neuron followed by a $softmax$ layer to normalize the output. The gambler's loss function for a single data point $(x, y)$ with $y$ being a $0 - 1$ label, and with neural network $f_\theta(\cdot) : X \to Y'$ is

$$\ell(f(x), y) = \log\left(f(x)_y + \frac{1}{o}f(x)_{m+1}\right) \tag{1}$$

$o$ is a hyperparameter for the loss function, and reasonable range of $o$ is $o \in (1, m]$ (also see ). Note that we omit $\theta$ when unnecessary, but $f$ always denote a neural network in this work, and should always come with subscript $\theta$, denoting its parameters. When training with gambler's loss, we notice that usually the following intuition hold. Larger $o$ encourages learning while smaller $o$ discourages learning. When $o$ is appropriate, lowering $o$ makes the neural network "cautious" about learning noises present in the dataset, and so lowering $o$ makes the model more robust to such noise. However, when $o$ is below a critical value, the minimum becomes to refrain from learning completely, and so achieving best performance using gambler's loss often involves approaching this critical value from above.

### 2.1 EARLY STOPPING CRITERION FOR THE GAMBLER'S LOSS

Now we propose a criterion to accurately estimate the location of the gap and perform early stopping at the peak in test accuracy. It is shown to be theoretically true that early stopping is effective against label noise in Li et al. (2019) for overparametrized networks trained with gradient descent. However, it is surprising that the existing literature only focuses on delaying or relieving the negative effect of learning the noisy labels, but not on finding the best time to do early-stop (Patrini et al., 2017; Han et al., 2018; Yu et al., 2019). As in the previous works Han et al. (2018); Yu et al. (2019), we assume that the $\epsilon$ is known (to perform early stopping); otherwise, it may be estimated.

Let $o$ be the gambler's hyperparameter. Let $\hat{p}$ be the predicted probability on the true label $y$, and let $\hat{k}$ denote the prediction made on all the wrong classes added altogether, $\hat{l}$ be the predicted confidence score by the gambler's loss. By definition of a probability distribution, we have $\hat{p} + \hat{k} + \hat{l} = 1$. Based on our observations, we make the following idealized assumption: during the the gap stage, the model has not learned anything about the data points with wrong labels, i.e., it makes strictly uniformly random guess across the all labels, the predicted probabilities are the same and are all equal to $\frac{\hat{k}}{m}$. This allows us to write down the general loss function during the gap stage:

$$\mathbb{E}[-\tilde{\ell}] = \epsilon \log\left(\hat{p} + \frac{\hat{k}}{m} + \frac{1 - \hat{p} - \hat{k}}{o}\right) + (1 - \epsilon)\log\left(\frac{\hat{k}}{m} + \frac{1 - \hat{p} - \hat{k}}{o}\right) \tag{2}$$

where the expectation is taken over the dataset, and we will omit it since it does not affect clarity. The following theorem uses the above assumptions and predict the loss of the gap stage.

**Theorem 1.** *During the idealized gap stage, the optimal solution has $\hat{k} = 0$, and so the loss function takes the form:*

$$-\tilde{\ell}(\hat{p}) = \epsilon \log\left(\hat{p} + \frac{1 - \hat{p}}{o}\right) + (1 - \epsilon)\log\left(\frac{1 - \hat{p}}{o}\right) \tag{3}$$

*which exhibits an optimal solution at*

$$-\tilde{\ell}^*(\epsilon, o) = \min_p -\tilde{\ell}(p) = \epsilon \log \epsilon + (1 - \epsilon)\log\left(\frac{1 - \epsilon}{o - 1}\right), \tag{4}$$

---

**Algorithm 1** Early Stopping on the Gambler's Loss

---

1: Given: untrained classifier parameters $\theta$, noise rate $\epsilon$, gambler's hyperparamter $o$.
2: Change classifier output from $m$-way to $(m + 1)$-way softmax.
3: **for** $(x, y)$ in each batch **do**
4:     Compute gambler's loss $\mathcal{L}_g$ from equation 1.
5:     If $\mathcal{L}_g < -\tilde{\ell}^*(\epsilon, o)$, end training.
6:     Update parameters $\theta$ using $\nabla_\theta \mathcal{L}_g$.
7: **end for**

---

*which only depends on $\epsilon$ and $o$, and we have $p^* = \frac{\epsilon o - 1}{o - 1}$.*

The proof is given in the appendix. This theorem tells us that a network trained with the gambler's loss function with hyperparameter $o$ on a symmetrically corrupted dataset with corruption rate $1 - \epsilon$ should have training loss around $-\tilde{\ell}^*(\epsilon, o)$ when entering the gap stage. This motivates using $-\tilde{\ell}^*(\epsilon, o)$ as the early stopping criterion. Notice that we expect our loss to have the following limits:

1. As $\epsilon \to 1$, we recover the no-noise setting, and the optimal loss is 0.

2. As $\epsilon \to 0$, we go to the full noise case, and the dis-confidence score would dominate, and the gambler's loss converge to $\log \frac{1}{o}$ (when $o$ is large) (Ziyin et al., 2019). We summarize our proposed method in Algorithm 1.

## 2.2 ROBUSTNESS TO LABEL NOISE WITHOUT EARLY STOPPING

One common approach (among others) to label noise involves introducing a surrogate loss function that is specialized for the corrupted dataset at hand and is not very useful when one is not aware of the existence of label noise. Therefore, one surprising finding in this work is that simply using the gambler's loss improves the performance of the model at convergence, thus provides robustness to label noise automatically. For example, in Figure 4a, we see that, without knowing $\epsilon$, the gambler's improves over the baseline by over $40\%$ testing accuracy at convergence.

We hypothesize that the reason is that the gambler's loss function makes learning on corrupted points slower than the clean points. As a result, this widens the test accuracy plateau by slowing down the memorization phenomenon. Let a data points be $x$, whose "observable" label is $y$, and true label is $y'$. Intuitively speaking, the mechanism is that it is easier to learn to reserve (by increasing $f(x)_{m+1}$) than to memorize the noisy points (by increasing $f(x)_y$), and so $f(x)_{m+1}$ saturates the $softmax$ layer, making the gradient vanish. Also, $o$ cannot be too small, since that will also make the gradient vanish on the clean data points.

To be slightly more formal. Notice that the gambler's loss function has derivative (defining $g(x)_y := f(x)_y + \frac{1}{o} f(x)_{m+1}$):

$$\nabla_\theta \log g(x)_y = \frac{1}{f(x)_y + \frac{1}{o} f(x)_{m+1}} \nabla g(x)_y \qquad (5)$$

and by Theorem 1 (in Section 2.1), during the gap stage we have:

$$\frac{1}{f(x)_y + \frac{1}{o} f(x)_{m+1}} = \begin{cases} \frac{1}{\epsilon}, & \text{if } y = y' \\ \frac{o-1}{1-\epsilon}, & y \neq y' \end{cases} \qquad (6)$$

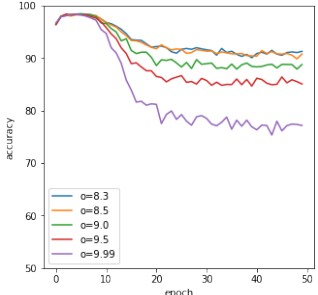

(a) Training with Adam

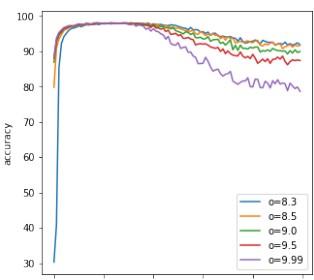

(b) Training with SGD

Figure 2: Training trajectories on the gambler's loss using different optimizers, with $r = 0.5$.

and the probability we update a clean data point is $\epsilon$, while that of a corrupted data is $\frac{1-\epsilon}{m-1}$, and, therefore, in expectation (sloppy notation), the gradient is of order

$$\mathbb{E}[\nabla \log g(x)_y] \sim \nabla g(x)_{y=y'} + \frac{o-1}{m-1} \nabla g(x)_{y \neq y'}, \qquad (7)$$

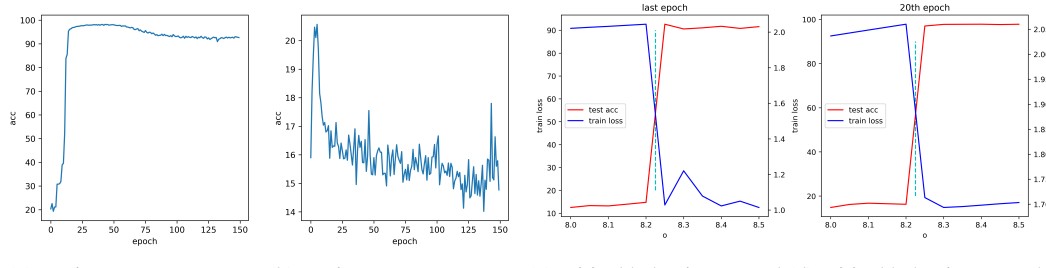

(a) testing accuracy at $o$ = 8.25 (b) testing accuracy at $o$ = 8.20 (c) critical behavior around $o$ = 8.2 at $epoch$ = 20 (d) critical behavior around $o$ = 8.2 at $epoch$ = 200

Figure 3: Critical behavior of the gambler's loss. Data showing that the learning almost do not happen at all for $o < o_{\text{crit}}$, while above $o_{\text{crit}}$ the behavior is qualitatively similar. The optimal $o$ can be tuned for using this phenomenon; however, more often one does not need to tune for o.

In general, $o - 1 < m - 1$ and so a corrupt data, in expectation, has gradient $\frac{o-1}{m-1}$ times smaller than that of the clean data, and, within a proper range of $o$, the lower the $o$ is, the slower the model learns the corrupt labels, providing stronger robustness (in fact, one might go one step further with this qualitative analysis, see section B). This trend is what we observe in Section 4.1 when $o$ lies above a critical value. Moreover, notice that this is independent of $\epsilon$, and we expect the gambler's loss to help even when $\epsilon$ is unknown, and this is shown in section 4.1. On the other hand, this analysis suggests that, if we train the gambler's loss with a subgradient methods such as Adam (Kingma & Ba, 2014), the constant factor in front of the gradient would be canceled by the preconditioner $1/\sqrt{\mathbb{E}g^2}$, and so the widening effect would not be observed. See Figure 2. The time it takes for Adam to reach and to start exiting the plateau seem invariant to $o$, while using SGD, we see that the plateau is widened as we decrease $o$. For Adam, even if the widening effect is not changed, we see that at convergence, smaller $o$ does lead to higher performance using both optimizers. More plots and experiments using Adam are given in the appendix. Moreover, as shown in Yu et al. (2019); Han et al. (2018), simply training on the raw loss functions features a gap stage that lasts only very shortly (~ 1 epoch), while the gambler's loss makes the network stays around its performance peak for a much longer time. The performance on the gambler's loss has barely dropped by the time the baseline almost dropped to random accuracy, and this an makes it easy to detect the peak and perform early-stopping.

### 2.3 TUNING FOR THE OPTIMAL $o$ AND FIRST-ORDER PHASE TRANSITION

We note that there is a "good" range for hyperparameter $o$ and a bad range. The good range is between a critical value $o_{\text{crit}}$ and $M$ (non-inclusive) and the bad range is smaller than $o_{\text{crit}}$. We conducted substantial experiments to observe that any $o$ within the good range are seen to provide improved robustness against label noise, and so most of the cases we do not tune for $o$ but only to make sure it is within the good range. Moreover, tuning for the optimal $o$ is fast and can be done in linear time, and we discuss this and its connection to the physical first-order phase transition theory in the appendix.

## 3 RELATED WORK

**Label Noise:** Modern dataset often contains a lot of labeling errors (Russakovsky et al., 2015; Schroff et al., 2010). Existing methods often focus on using a surrogate loss function (Patrini et al., 2017) that is specific to the label noise problem at hand, or design a special training scheme to alleviate the negative effect of learning the data points with wrong labels (Han et al., 2018; Yu et al., 2019). In this work, we mainly compare with the following methods. **F-Correction (FC)** Patrini et al. (2017): this method smooths the hard labels with the noise transition matrix; we estimate the transition matrix as suggested by the paper; **Co-teaching (CT)** (Han et al., 2018): this method trains two networks simultaneously, and update each with the other's predicted label to decouple the mistakes; **Co-teaching+ (CT+)** (Yu et al., 2019): this method improves the previous method by only propagating on the labels the two networks disagree. Also, we use the subscript **ES** as a shorthand for early stopping. Since the early stopping criterion for these methods are not clear, we early stop at where the training accuracy is larger than $\epsilon$ for comparison, since this is when the learning on the noisy data has definitely started.

**Early Stopping**. Early stopping is an old problem in machine learning Prechelt (1998); Amari (1998), and studying this in the setting of label noise also appeared recently Li et al. (2019); Hu et al. (2019). In particular, we propose to early stop on a loss function called the gambler's loss function Ziyin et al.

(2019). Our analysis on this loss function allows us to propose a very simple analytic function to predict a early stopping threshhold without using a validation set and is independent of the model (as long as it is a neural network with enough capacity) or the task. Early stopping without validation set is also recently studied, but does not relate to the label noise problem (Mahsereci et al., 2017). It has been long noticed (Frénay & Verleysen, 2013; Han et al., 2018) and recently theoretically proven (Li et al., 2019) that early stopping can an effective way to defend against label corruption (but no actual method or heuristic for early stopping is given); to the best of our knowledge, this is first early stopping criterion to deal with the problem.

## 4 EXPERIMENTS

We design several experiments to evaluate 1) the robustness of gambler's loss to noisy labels, 2) the effectiveness of our early stopping criterion, and 3) the performance of gambler's loss when applied to existing methods to combat noisy labels.

### 4.1 ROBUSTNESS OF GAMBLER'S LOSS TO LABEL NOISE

We first demonstrate that simple training on the gambler's loss provides robustness against label noise (also see Section 2.3). For this and later sections, the exact experimental details are given in the Appendix. For demonstration, we choose a simple CNN with 2 convolutional layers followed by 2 fully connected layers. See Figure 4. We see that with properly tuned $o$, the gamblers's loss alleviates the drop in performance caused by learning the corrupt labels. For example, in Figure 4a, all three choices of $o$ makes the accuracy plateau wider than training on the raw loss (nll loss). At convergence, the final accuracy is much higher (~ 90%) than the baseline (~ 55%). For the experiments on CIFAR10, the results are slightly subtler. Here we see the effect of not properly tuning $o$. As shown in (Ziyin et al., 2019), the proper choice ranges from 1 to $M$; higher $o$ ($\approx M$) encourages learning while very low $o$ (~ 1) slows down learning. We notice that for a different corruption rate, a different $o_{\text{crit}}$ seem to exist. Also see Table 1 (category $Gbler$), we see that at convergence, training on gambler's loss achieves much higher accuracy than the baseline training on *nll* loss.

### 4.2 EARLY STOPPING

Now, we show that stopping at our predicted loss level succeeds in stopping at a point close to where the maximum accuracy is achieved. See Figure 5. We choose architectures of very different capacity. The experiment on MNIST is done as in the previous section and the experiment on CIFAR-10 is done using ResNet-18. We also test on a wide range of noise levels ranging from small, with corruption rate $0.2$, to extremely large, with $r = 0.85$ (the intermediate ranges with larger figures are given in the appendix, see Figure 7). We see that our method predicts close to optimal early stopping point for all corruption rates we tested on. Also notice how the learning curves are different for the two different datasets. This suggests that our theory works even in the presence of very different dynamics. The fact that this early stopping criterion works suggests in return suggests the validity of our assumption. In these experiment, we do not tune for optimal $o$, they are all set to $o = 9.9$; since the results are qualitatively similar when $o > o_{\text{crit}}$.

See Table 1 and also compare with other baselines in Table 2. We notice that doing this already achieves SOTA results on MNIST for all symmetric noise categories according to the results in Han et al. (2018) and Yu et al. (2019). Early stopping (AES) using our criterion is compared with a very standard early stopping method using validation set: we split 6000 images from the training set to make a validation set, and we early stop when the validation accuracy stops to increase for 5 consecutive epochs. There are also a few other validation-based early stopping

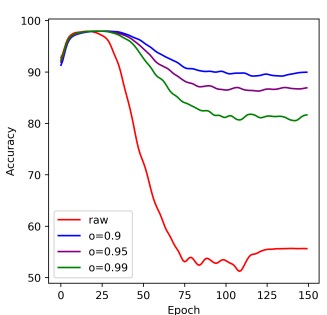

(a) MNIST: $r = 0.5$

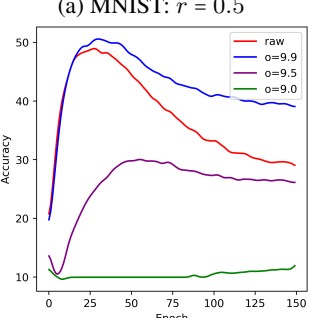

(b) CIFAR10: $r = 0.5$

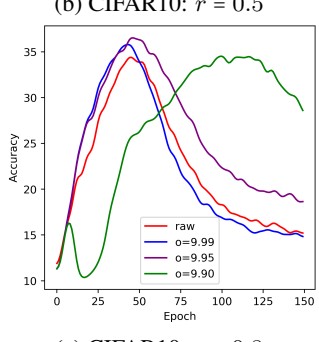

(c) CIFAR10: $r = 0.8$

Figure 4: Testing accuracy through out training; $raw$ refers training on the default loss function, i.e. the *nll* loss, $raw$ is plotted in red. We see that, when $o > o_{\text{crit}}$, the gambler's loss provides robustness on its against noisy labels.

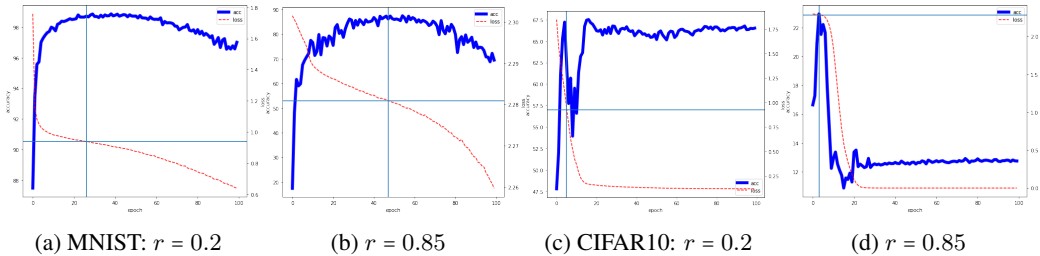

(a) MNIST: $r = 0.2$      (b) $r = 0.85$      (c) CIFAR10: $r = 0.2$      (d) $r = 0.85$

Figure 5: Early stopping on MNIST (1st row) and CIFAR10 (2nd row). $r$ refers to corruption rate. The horizontal line is the predicted early stopping point. We see that this point corresponds to where the testing accuracy (blue solid line) is at maximum.

Table 1: Robustness of the Gambler's loss to noisy labels. We see that simply using gambler's loss on label noise problems improve. For the two categories using early stopping, the number in parenthesis refers to the avg. number of epochs to stop. Using gambler's loss stops at better accuracy, has smaller variance, and takes much shorter time.

| **Dataset** | Performance Comparisons | | Early Stopping Comparisons | |
|---|---|---|---|---|
| | *nll* loss | Gblers | VES | Gblers$_{AES}$ |
| MN $r = 0.2$ | $84.7 \pm 0.5$ | $96.7 \pm 0.2$ | $95.1 \pm 0.4$ (95) | $\mathbf{98.8 \pm 0.1\ (17)}$ |
| MN $r = 0.5$ | $55.1 \pm 3.1$ | $91.2 \pm 0.7$ | $79.7 \pm 2.0$ (115) | $\mathbf{98.0 \pm 0.0\ (18)}$ |
| MN $r = 0.65$ | $39.7 \pm 2.5$ | $85.9 \pm 1.1$ | $58.9 \pm 2.5$ (115) | $\mathbf{97.2 \pm 0.2\ (15)}$ |
| MN $r = 0.8$ | $19.1 \pm 3.0$ | $76.1 \pm 0.3$ | $21.1 \pm 0.1$ (117) | $\mathbf{93.5 \pm 0.3\ (15)}$ |
| MN $r = 0.85$ | $14.5 \pm 0.7$ | $71.0 \pm 1.4$ | $15.0 \pm 1.1$ (110) | $\mathbf{85.2 \pm 1.7\ (13)}$ |
| IMDB $r = 0.1$ | $70.3 \pm 0.5$ | $70.5 \pm 0.3$ | $71.0 \pm 0.5$ (46) | $\mathbf{74.0 \pm 0.4(10)}$ |
| IMDB $r = 0.15$ | $64.3 \pm 0.3$ | $65.6 \pm 0.2$ | $67.0 \pm 0.3$ (21) | $\mathbf{69.1 \pm 0.1\ (12)}$ |
| IMDB $r = 0.2$ | $58.7 \pm 0.1$ | $60.9 \pm 0.3$ | $59.6 \pm 0.8(8)$ | $\mathbf{65.3 \pm 0.7(12)}$ |
| IMDB $r = 0.25$ | $56.4 \pm 0.2$ | $58.3 \pm 0.1$ | $55.6 \pm 0.2$ (8) | $\mathbf{62.4 \pm 0.4(9)}$ |
| IMDB $r = 0.3$ | $51.8 \pm 0.3$ | $53.8 \pm 0.4$ | $51.1 \pm 0.2$ (8) | $\mathbf{58.2 \pm 0.3(12)}$ |

criterion, but they are shown to perform qualitatively and quantitatively similar (Prechelt, 1998), so we only compare with this method. This early stopping is called **VES** (validation early stopping) and our method is called **AES** (analytical early stopping). We also introduce a warmup technique for **AES**, where we set $o = m$ for the first five epochs; this technieques demonstrates consistent gain throughout, and we denote this as **LAES**. We see that our proposed method significantly outperforms the baseline early stopping method; this is because a small validation set may have large variance and this problem is more serious when label noise is present (since validation sets are often split from the train set). In the VES vs. AES comparison, we fix $o = 9.99$ and is not directly comparable to the left parts in the table. In the left part, We also notice that training and stopping on the gambler's loss is especially effective on tasks with extreme corruption rate ($r = 0.85$), with around $70\%$ accuracy improvement over the baseline and $28\%$ improvement over the runner-up SOTA methods. We also conduct experiment on the IMDB dataset (Maas et al., 2011), which is a standard NLP task on sentiment analysis and is a binary classification problem. We use a standard LSTM with 256 hidden dimension, and with 300-dimensional pretrained GloVe word embeddings (Pennington et al., 2014). Again, we notice that, 1), using the gambler's loss consistently improve on the *nll* loss, and 2) using AES consistently improves on early stopping on a validation set (by about $2 - 7\%$ in absolute accuracy). Also see appendix section M for experiments on really small corruption rate $r = 0.02 - 0.08$ on IMDB.

### 4.3 BENCHMARKING AGAINST OTHER METHODS

In this section, we mainly use **LAES** as the proposed method, since it consistently outperforms **AES** (for example, see the CIFAR-10 experiments, and see section L). See table 2. On MNIST, using our method always achieves SOTA performance. On CIFAR-10 (CF10), for the CIFAR10 experiments, we do not compare with CT+ since it is a modified version of CT and we notice that it fails in the same range as CT ($r = 0.7, 0.8, 0.85$). Our method is especially strong when the noise level is extreme ($r \sim 0.7, 0.8, 0.85$), agreeing with our finding in the previous section. We also note that LAES consistently outperforms AES. CT seems to outperform our method a little in the range $r \sim 0.1 - 0.6$, but CT requires twice as much parameters, and takes at least twice as much time to train due to its special training scheme. We also emphasize that the improvement over purely training on the *nll* loss is significant, ranging from $10 - 30\%$ in absolute accuracy. We note that we also tried to

Table 2: On MNIST, using our method always achieves SOTA performance. On CIFAR-10 (CF10), our method is especially strong when the noise level is extreme ($r \sim 0.7, 0.8, 0.85$), agreeing with our finding in the previous section. (*NA means that the ES stopping point is not reached by the algorithm.)

| Dataset | FC | CT | CT+ | $CT_{ES}$ | $CT+_{ES}$ | $Gbler_{LAES}$ |
|---|---|---|---|---|---|---|
| MN $r = 0.2$ | $98.8 \pm 0.1$ | $97.2 \pm 0.0$ | $97.8 \pm 0.1$ | $98.6 \pm 0.2$ | $97.6 \pm 0.1$ | $\mathbf{99.0 \pm 0.1}$ |
| MN $r = 0.5$ | $79.6 \pm 2.0$ | $93.3 \pm 0.1$ | $95.4 \pm 0.2$ | $97.6 \pm 0.1$ | $96.3 \pm 0.2$ | $\mathbf{98.4 \pm 0.2}$ |
| MN $r = 0.65$ | $29.1 \pm 0.1$ | $92.5 \pm 0.1$ | $92.5 \pm 0.1$ | $\mathbf{97.3 \pm 0.2}$ | $95.6 \pm 0.2$ | $\mathbf{97.6 \pm 0.3}$ |
| MN $r = 0.8$ | $19.8 \pm 0.1$ | $78.1 \pm 0.5$ | $67.5 \pm 0.1$ | $82.1 \pm 1.7$ | $71.4 \pm 0.3$ | $\mathbf{95.0 \pm 0.5}$ |
| MN $r = 0.85$ | $11.8 \pm 0.0$ | $60.5 \pm 0.2$ | $10.1 \pm 0.0$ | $66.5 \pm 0.5$ | $NA$ | $\mathbf{88.0 \pm 1.0}$ |
| | *nll* loss | CT | $CT_{ES}$ | $Gblrs_{AES}$ | $Gblrs_{LAES}$ | |
| CF10 $r = 0.2$ | $58.1 \pm 0.1$ | $\mathbf{71.0 \pm 0.6}$ | $69.5 \pm 0.4$ | $67.5 \pm 1.0$ | $68.2 \pm 0.0$ | |
| CF10 $r = 0.5$ | $35.4 \pm 0.2$ | $\mathbf{65.2 \pm 0.2}$ | $64.5 \pm 0.8$ | $59.5 \pm 0.4$ | $60.1 \pm 0.2$ | |
| CF10 $r = 0.6$ | $28.3 \pm 0.1$ | $61.1 \pm 0.2$ | $\mathbf{62.0 \pm 0.3}$ | $51.3 \pm 0.5$ | $57.1 \pm 0.5$ | |
| CF10 $r = 0.7$ | $20.9 \pm 0.4$ | $48.9 \pm 0.2$ | $49.1 \pm 1.0$ | $48.2 \pm 0.5$ | $\mathbf{50.6 \pm 0.5}$ | |
| CF10 $r = 0.8$ | $15.1 \pm 0.5$ | $25.0 \pm 0.1$ | $25.8 \pm 1.0$ | $37.0 \pm 0.7$ | $\mathbf{40.7 \pm 2.0}$ | |
| CF10 $r = 0.85$ | $13.0 \pm 0.5$ | $17.3 \pm 0.1$ | $NA$ | $18.9 \pm 1.0$ | $\mathbf{27.0 \pm 2.1}$ | |

combine our method with CT but was not very successful, but CT throw away a large portion of the training dataset, and changes the underlying distribution, making the AES criterion not applicable.

## 5 CONCLUSION AND DISCUSSION

In this work, we have proposed a simple and effective method to deal with label noise problems. This paper promotes the gambler's loss as a noise robust loss function, shown to improve performance of existing classifier model, while the early stopping criterion provides a simple criterion for early stopping when the corruption rate is known. Depending on the problem, one may decide to use the complete framework together (gambler's loss training + early stopping), or, when the assumption is not well satisfied, only use the gambler's loss to improve the model's robustness against a label-corrupted learning task. This urges for an understanding for why the gambler's loss actually works. We plot in training accuracy for MNIST with corruption rate $0.8$ in Figure 6. We see that the gambler's loss seems to work in a fundamentally different way from the standard *nll* loss. Training on *nll* loss achieves $100\%$ accuracy on the training set, with its loss approaching $0$; however, the gambler's loss does not do so, its training accuracy stops increasing at around $36\%$, which is slightly above $\epsilon$. We note that this observation also holds for various corruption rates, and for different optimizers such as SGD. Comparing with Table 5, we see that the gambler's loss achieves about $76\%$ testing accuracy, while the baseline drops to $19\%$ at this time. The vertical line also shows where our early stopping criterion is reached, at this point is training accuracy is $\sim 20\%$ with testing accuracy $\sim 92\%$. We hypothesize that the main reason for gambler's loss's outperforming the baselines is that it automatically chooses a subset of the training set to learn on, thus avoiding a large part of the dataset that is corrupted.

This result may provide insight into the current debate on the generalization-memorization effect (Zhang et al., 2017; Wu et al., 2017; Hu et al., 2019; Arpit et al., 2017). A key argument from Zhang et al. (2017) was "*deep neural networks easily fit random labels...; explicit regularization may improve generalization performance, but is neither necessary nor by itself sufficient for controlling generalization error.*" Our results above, however, shows that this is not the case when a different loss function is chosen. While the *nll* loss encourages learning and memorizing the training set, the gambler's loss does not seem to do so. Our findings suggests that the key to understanding the memorization effect might be on the loss function itself, not the architecture of neural networks or explicit regularization; this is also what is suggested by Abiodun et al. (2018). We hypothesize that it may be seen as a natural generalization of the *nll* loss function, since it only involves adding a task independent term to the original *nll* loss, and using $o > o_{crit}$ seems always beneficial. We would like to study the memorization effect using the gambler's loss in more detail. Applications to real-world large-scale datasets with real label noise are also ripe avenues for future work.

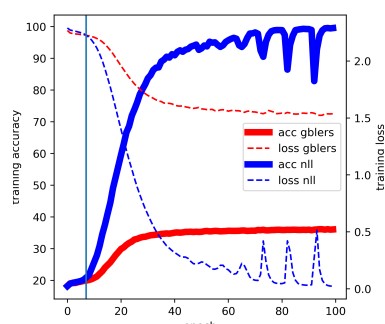

Figure 6: Training accuracy and training loss On MNIST with corruption rate $0.8$ with Adam. $o = 9.7$. Training with gambler's loss prevents memorization of noisy labels. At convergence, *nll* loss reaches $19\%$ testing accuracy, while the gambler's loss stays around $76\%$.

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

## A   A BRIEF INTRODUCTION TO THE GAMBLER'S LOSS

Let $f(\cdot)$ denote the model and $f(x)$ is the $m+1$-dimensional normalized prediction, i.e.,

$$\sum_{k=1}^{m+1} f(x)_k = 1. \tag{8}$$

The gambler's loss for a classification problem takes the form

$$\max \ell(f(x), y) = \sum_{k=1}^{m+1} p_{k,y} \log\left(f(x)_k + \frac{1}{o} f(x)_{m+1}\right) \tag{9}$$

where $p_{k,x} = Pr(k|x)$ is the true label. Notice that if $o \to \infty$, the loss function is equivalent to the KL divergence. The empirical target is usually a $1 - 0$ loss, where $p_{y,x} = 1$ for the true label $y$ and zero for $k \neq y$; thus the loss function takes the following simple form (and we change $\max \ell$ to $\min -\ell$):

$$\min -\ell(f(x), y) = -\sum_{k=1}^{m+1} \delta_{k,y} \log\left(f(x)_k + \frac{1}{o} f(x)_{m+1}\right) \tag{10}$$

$$= -\log\left(f(x)_y + \frac{1}{o} f(x)_{m+1}\right). \tag{11}$$

and this reduces to the cross entropy (or nll loss) if $o \to \infty$.

It is also useful to notice the following relation. Let $\hat{p}$ be the predicted probability on the true label $y$, and let $\hat{k}$ denote the prediction made on all the wrong classes added altogether, $\hat{l}$ be the predicted confidence score by the gambler's loss. To relate this to the definition above (Equation 11), we note that this is equivalent to defining $\hat{p} := f(x)_y$, $\hat{k} := \sum_{i \neq y}^{m} f(x)_i$, $\hat{l} := f(x)_{m+1}$, and, therefore, by the normalization condition in Equation 8

$$\hat{p} + \hat{k} + \hat{l} = f(x)_y + \sum_{i \neq y}^{m} f(x)_i + f(x)_{m+1} = 1. \tag{12}$$

and this condition will be used to prove Theorem 1.

## B   A TRAINING SCHEDULE

Recall equation 7:

$$\mathbb{E}[\nabla \log g(x)_y] \sim \nabla g(x)_{y=y'} + \frac{o-1}{m-1} \nabla g(x)_{y \neq y'} \tag{13}$$

We can go one step further and plug in $g(x)_y = f(x)_y + \frac{1}{o} f(x)_{m+1}$ to consider the effect on the prediction for $f(x)_{j=y'}, f(x)_{j \neq y', j < m+1}$, and $f(x)_{m+1}$. If $|\nabla f(x)_{m+1}| > |\nabla f(x)_{j \neq m+1}|$, we say that the reseravation score $f(x)_{m+1}$ is prone to saturation, and learning on the label class $j$ is prevented by such saturation. We obtain:

$$\mathbb{E}[\nabla \log g(x)_y] \sim \nabla f(x)_{j=y'} + \frac{o-1}{m-1} \nabla f(x)_{j \neq y', j < m+1} + \left(\frac{1}{o} + \frac{o-1}{o(m-1)}\right) \nabla f(x)_{m+1} \tag{14}$$

where the first term is the gradient for the correct label, and the second term he gradient for the incorrect labels and the third term is the gradient for the reservation score. Roughly estimating, to prevent saturation delaying learning on correct labels, we want $1 > \frac{1}{o} + \frac{o-1}{o(m-1)}$, for MNIST and CIFAR 10 we have $m = 10$, and this condition translates to $o > 8.88...$, which agree in magnitude to the experimental threshold $o_{crit} \approx 8.2$. This high-level agreement suggests that our qualitative analysis is good.

On the other hand, if we want the saturation on $f(x)_{m+1}$ to prevent learning on the corrupt labels completely, then we should have $\frac{1}{o} + \frac{o-1}{o(m-1)} > \frac{o-1}{m-1}$, and for $m = 10$, this translates to $o < 4$. This suggests the following training scheme for $m = 10$ problems:

1. start training with $o > 8.89$
2. when the early stopping criterion is met, either stop, or switch to $o < 4$

In fact, similar training scheme is used in (Ziyin et al., 2019) in order to achieve SOTA results, while unaware of this analysis. For CIFAR10, they started with $o = 9.9$ to for the first fifty epochs, and then switched to $o = 2$ afterwards to achieve the best performance.

## C    MATHEMATICAL SYMBOLS

$$N : \ number \ of \ input \ dimension$$

$$M : \ number \ of \ output \ dimension$$

$$\mathcal{D} : \ dataset, \ |\mathcal{D}| : \ size \ of \ the \ dataset$$

## D    EXPERIMENT DETAILS

We use Pytorch as the framework of implementation. Code to this paper will be released at *https://\*\*\*\*\*\*\*\**

## E    THEOREM PROOF

In this section we prove theorem 1. We first show that $\hat{k} = 0$. Intuitively speaking, this simply means that a gambler betting randomly will not make money, and so the better strategy is to reserve money in the pocket, and so it suffices to show that for any solution $\hat{p}, \hat{k}, \hat{l}$, the solution $\hat{p}' = \hat{p}, \hat{k}' = 0, \hat{l}' = \hat{l} + \hat{k}$ achieves better or equal doubling rate. For a mislabeled point (we drop $\hat{\cdot}$), the loss is $\log(\frac{k}{M} + \frac{l}{o})$ but $M > o$, and so $\log(\frac{k}{M} + \frac{l}{o}) < \log(\frac{k+l}{o})$, and we have that optimal solution always have $\hat{k} = 0$.

Now, we find the optimal solution to

$$- \tilde{\ell}(\hat{p}) = \epsilon \log \left( \hat{p} + \frac{1 - \hat{p}}{o} \right) + (1 - \epsilon) \log \left( \frac{1 - \hat{p}}{o} \right) \tag{15}$$

by taking the derivative with respect to p:

$$- \frac{\partial \tilde{\ell}}{\hat{p}}(\hat{p}) = \epsilon \frac{o - 1}{(o - 1)\hat{p} + 1} + (1 - \epsilon)\frac{-1}{1 - \hat{p}} \tag{16}$$

and then setting it equal to 0

$$- \frac{\partial \tilde{\ell}}{\hat{p}}(\hat{p}) = \epsilon \frac{o - 1}{(o - 1)\hat{p} + 1} + (1 - \epsilon)\frac{-1}{1 - \hat{p}} = 0 \tag{17}$$

is the $\hat{p}_{optimal}$:

$$\hat{p}_{optimal} = \frac{\epsilon o - 1}{o - 1} \tag{18}$$

and then plugging into the original equation [9]:

$$- \tilde{\ell}^*(\epsilon, o) = \min_{p} -\tilde{\ell}(p) = \epsilon \log \epsilon + (1 - \epsilon) \log \left( \frac{1 - \epsilon}{o - 1} \right) \tag{19}$$

## F    ADDITIONAL EXPERIMENTS

In this section, we give more experiment results on Gambler + Early Stopping. See Figure 7

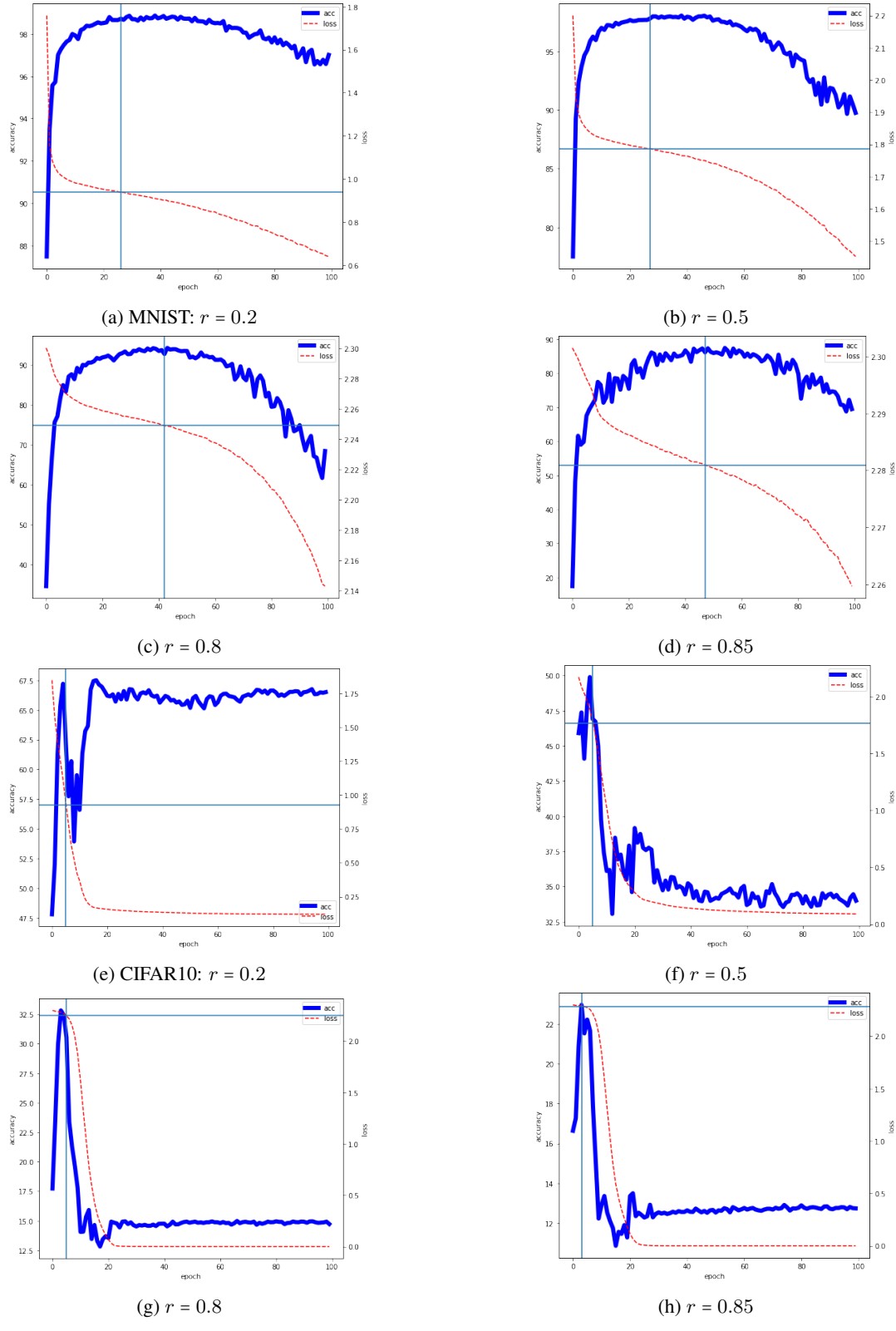

Figure 7: Early stopping on MNIST (1st and 2nd row) and CIFAR10 (3rd and 4th row). $r$ refers to corruption rate. The horizontal line is the predicted early stopping point. We see that this point almost always corresponds to where the testing accuracy (vertical blue solid line) is at maximum.

## G    DEFINITION OF TRANSITION MATRIX

Transition matrix is used to replace some correct labels with wrong labels because the dataset used in our experiments such as MNIST and CIFAR are clean. There are two different types of noise transition matrix and explanation follows.

**Symmetric Flipping**: Each label has an uniform possibility to transform to one of the rest class. In most cases of manual label work, a correct label has equal possibility to be any other class label. n is the number of class and $\delta$ is the rate of label that will be modified as a wrong label

$$M := \begin{bmatrix} 1-\delta & \frac{\delta}{1-n} & \frac{\delta}{1-n} & \cdots & \frac{\delta}{1-n} \\ \frac{\delta}{1-n} & 1-\delta & \frac{\delta}{1-n} & \cdots & \frac{\delta}{1-n} \\ \frac{\delta}{1-n} & \frac{\delta}{1-n} & 1-\delta & \cdots & \frac{\delta}{1-n} \\ \cdots & \cdots & \cdots & \cdots & \cdots \\ \frac{\delta}{1-n} & \frac{\delta}{1-n} & \frac{\delta}{1-n} & \cdots & 1-\delta \end{bmatrix} \tag{20}$$

**Pair Flipping**: Each class can only be corrupted as a specific class. For example, it's more likely to label 6 as 9 than any other class except 9. $\delta$ is the rate of label that will be modified as a wrong label:

$$M := \begin{bmatrix} 1-\delta & \delta & 0 & \cdots & 0 \\ 0 & 1-\delta & \delta & \cdots & 0 \\ 0 & 0 & 1-\delta & \cdots & 0 \\ \vdots & \vdots & \vdots & \vdots & \vdots \\ \delta & 0 & 0 & \cdots & 1-\delta \end{bmatrix} \tag{21}$$

## H    CNN ARCHITECTURE

In this section, we show the architectures that we used but did not describe in the main text.

Table 3: architecture of the neural network used in MNIST

| CCNN on MNIST |
| --- |
| $28 \times 28$ Gray Image |
| $20 \times 5$ conv, 1 ReLU |
| $2 \times 2$ max-pool, stride 2 |
| $50 \times 5$ conv, 20 ReLU |
| $2 \times 2$ max-pool, stride 2 |
| dense $800 \rightarrow 500$ |
| dense $500 \rightarrow 11$ |

Table 4: architecture of the neural network used in cifar10

| CNN on cifar10 |
| --- |
| $32 \times 32$ RGB Image |
| $5 \times 5$ conv, 128 LReLU |
| $5 \times 5$ conv, 128 LReLU |
| $2 \times 2$ max-pool, stride 2 |
| dense $128 \rightarrow 128$ |
| dense $128 \rightarrow 11$ |

## I    EFFECT OF TUNING O AND CRITICAL BEHAVIOR

In this section, we give more experiments on the critical behavior discussed in 2.3. First see Figure 8.

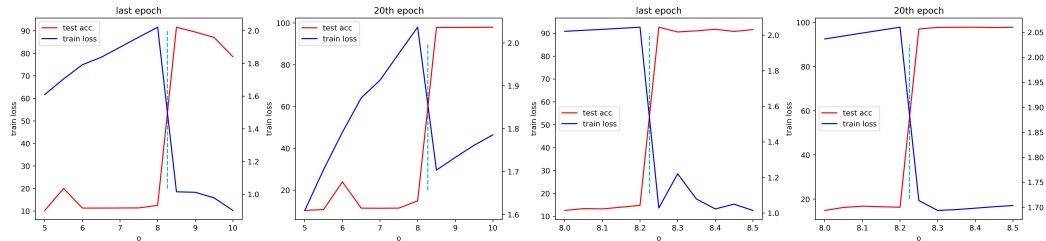

(a) critical behavior around (b) critical behavior around (c) critical behavior around (d) critical behavior around
$o = 8.2$ at $epoch = 200$ $\quad$ $o = 8.2$ at $epoch = 200$ $\quad$ $o = 8.2$ at $epoch = 20$ $\quad$ $o = 8.2$ at $epoch = 200$

Figure 8: critical behavior of the gambler's loss. We see that the learning almost do not happen at all for $o < o_{crit}$, while above $o_{crit}$ the behavior is qualitatively similar. The optimal $o$ can be tuned for using this phenomenon; however, more often one does not need to tune for o.

In Figure 2, we show that lower value of $o$ provides stronger robustness to label noise consistently; this raises the question about whether we can continue to lower $o$ to its lower limit (= 1) to achieve the best performance or not - the answer is no, but the reason is very surprising. We empirically find that a first order phase transition exists when tuning $o$ (such as transition of water to ice). In particular, we find that there exists a critical value $o_{\mathrm{crit}}$ such that, when $o > o_{\mathrm{crit}}$, the robustness to label noise increases as we lower $o$; when $o < o_{\mathrm{crit}}$, the learning proceeds badly: *the change in performance is discontinuous*. This is characteristic of a physical first order phase transition (Landau & Lifshitz, 2013). See Figure 3. In this setting, we have $r = 0.5$ for MNIST, and we pin down the $o_{\mathrm{crit}}$ to lie between $8.21$ and $8.22$; qualitatively different behaviors are found for the two different phase. See appendix for plots ranging from $o = 10.0$ to $o = 5.0$. A qualitatively similar behavior is observed using Adam[1] and on CIFAR10, and using other architectures. Clearly studying this critical behavior is very interesting but beyond the scope of this work. The above observation, however, suggests the following heuristic to tune hyperparameter $o$:

1. Start with $o$ very close to $M$ (for reference, for MNIST, $o \approx 9.5$ seems a good starting point; for CIFAR10, $o \approx 9.9$; CIFAR100: $o \approx 99.9$).

2. If the learning curve starts similarly as before, then decrease $o$ until the learning stops abruptly, and use the previous $o$

This procedure for tuning $o$ does not rely on a validation set at all. Also, as show in in Figure 2 for $o = 8.3, 8.5$, the performance is very similar when $o$ is close to $o_{\mathrm{crit}}$, there is no need to be infinitesimally close to the critical value. Unless otherwise noted, we tune our hyperparameter this way in the experiment section. Moreover, One can prove that $o = M$ will always work at least as good as the baseline, and so $o_{\mathrm{cr}}$ should always $< M$.

---

[1]Using Adam we can even pin down $o_{\mathrm{crit}}$ for this problem to lie between $8.2100$ and $8.2101$

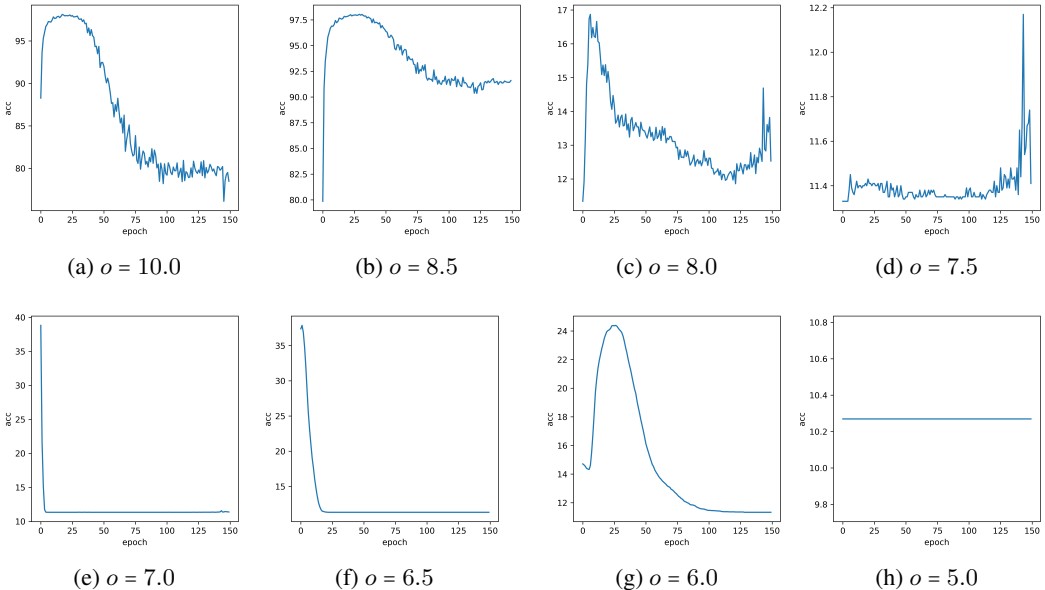

Figure 9: Gambler's loss is used alone without early stopping method. Initially, the performance gets better with the $o$ decreasing. While the performance gets worse with the $o$ decreasing when $o$ is less than 8.5, which indicates there is a critical point for $o$ locates in between 8.0 and 8.5. The most suitable $o$ is just larger than the critical point

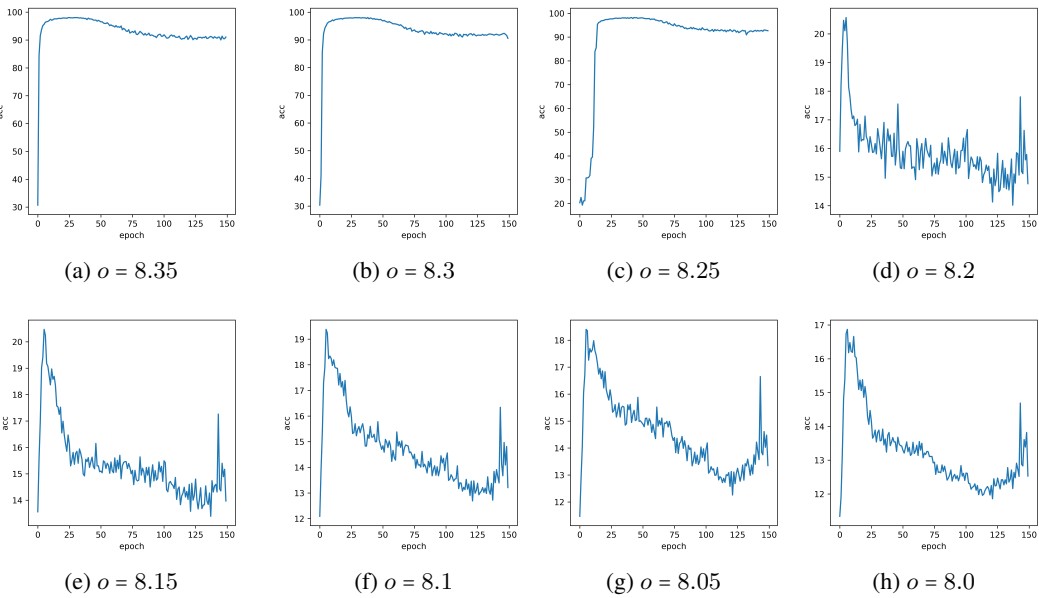

Figure 10: Under higher resolution on hyperparameter $o$, it's for sure that the performance will get better when $o$ gets closer to critical point from right side on number axis. While the performance will dramatically get terrible when $o$ is just lower than the critical points. What's more, the critical point is in the range of $[8.2, 8.25]$

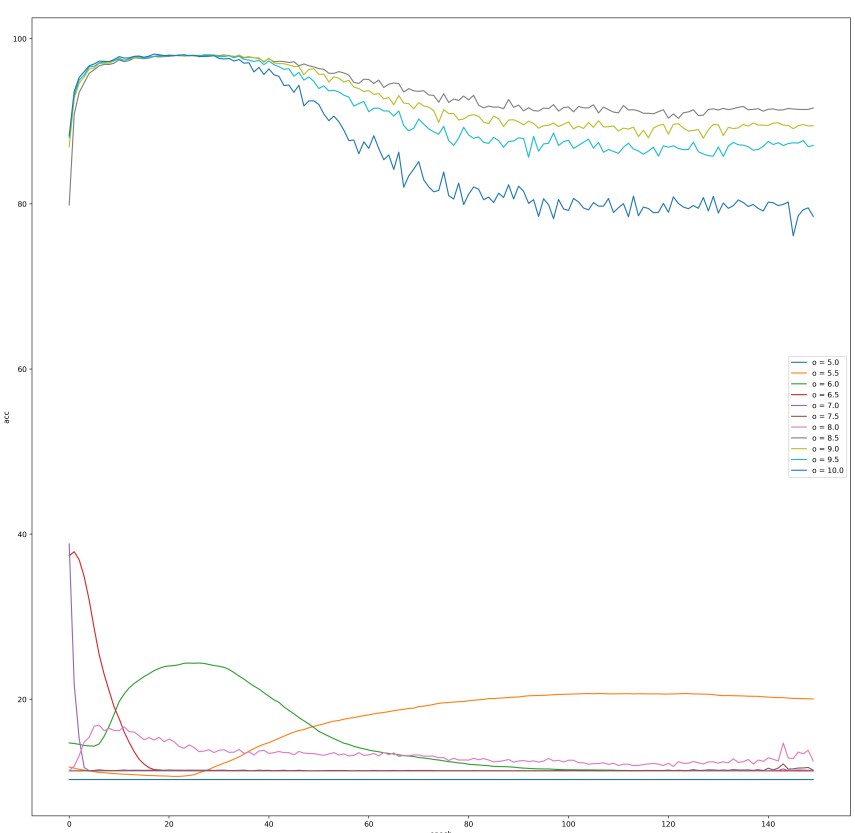

Figure 11: put all curve of Figure 8 in one Figure

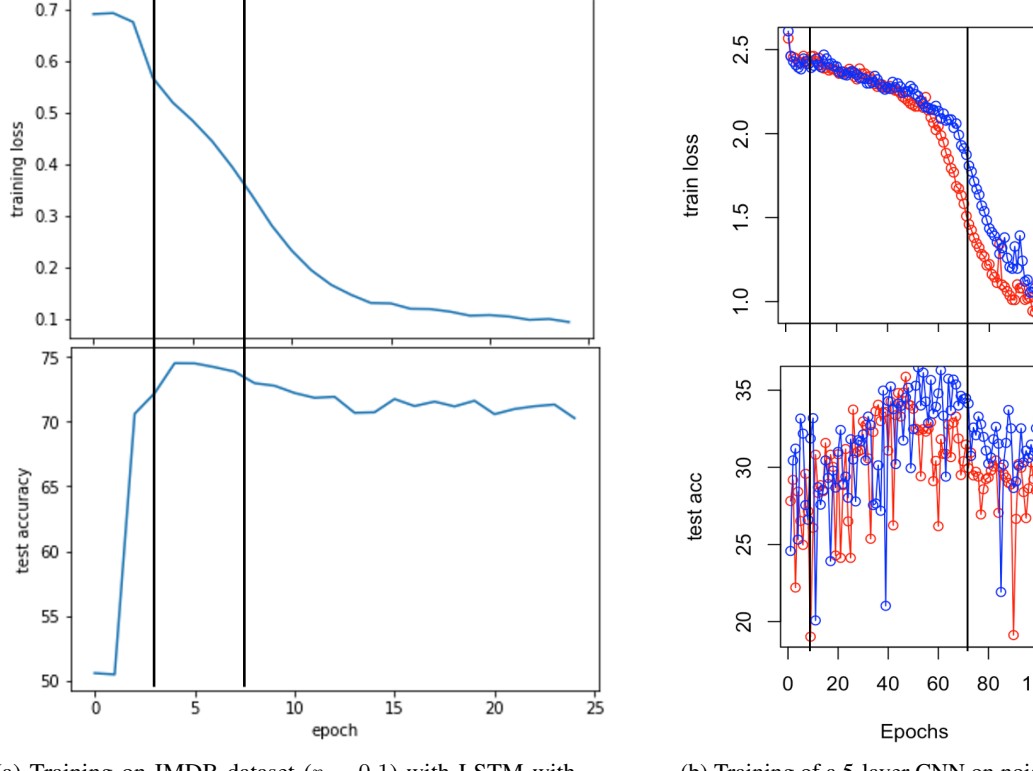

(a) Training on IMDB dataset ($r = 0.1$) with LSTM with pretrained GloVe embedding.

(b) Training of a 5-layer CNN on noisy openimage dataset ($r$ unknown).

Figure 12: Three stage phenomenon across different datasets and architectures

## J    MORE EVIDENCE ON THE THREE STAGE PHENOMENON

The three stage learning curve observed in section 1 is actually quite universal when significant level of label noise is present. Of course, the exact details of the curve really depends on the architecures and the tasks. See Figure 12. The left pictures shows training of an LSTM with hidden size 300 with pretrained GloVe embedding (Pennington et al., 2014) on the IMDB dataset (Maas et al., 2011). Notice that the performance peak (around epoch 5) also coincides with the plateau, as we expected.

On the right, we try a 5-layer CNN on the openimage dataset (Krasin et al., 2016). Two lines are two independent runs. This is a very noisy dataset with many mis-images that do not belong to any category in the dataset. In this dataset, the noisy are not artificially introduced by flipping or symmetric corrupt, and so the corruption rate is not known a priori. However, we still see the three stage phenomenon due to the existence of strong label noise in the dataset. The plauteau is wide and easily observable in this case, also notice that the end of plateau corresponds to the performance peak. Moreover, this plot actually shows one more advantage of the our method: by observing the learning curve by naked eyes, one identify that the end of plateau occurs at loss ~ 2.2, and plug this back into our early stopping criterion, one may estimate the corruption rate to be roughly $0.6 - 0.7$.

## K    SOME EXPERIMENTS ON ASYMMETRIC NOISES

In this section, we present some experiments on asymmetric noise. In particular, here we consider the pairflip kind of asymmetric noise. The definition is given in sectionG, which is the same as was given in Han et al. (2018). See Table 5. While using gambler's loss seems to always improve performance at convergence, performing early stopping is tricky in this case, since the criterion in Theorem 1 relies on the fact that the labels are symmetrically corrupted. In this sense, in order to perform early stopping, we notice that the problem has an effective dimension of 2 (because the

Table 5: Robustness of the Gambler's loss to asymetric noise. We set $o = 2.0$ for this experiment.

| Dataset | $nll$ loss | Gblers | Gblers$_{LAES}$ |
|---|---|---|---|
| MN pairflip $r = 0.3$ | $71.8 \pm 0.5$ | $74.7 \pm 0.8$ | $96.3 \pm 1.5$ |
| MN pairflip $r = 0.4$ | $59.6 \pm 1.0$ | $63.0 \pm 1.0$ | $90.6 \pm 2.0$ |
| MN pairflip $r = 0.45$ | $55.2 \pm 2.1$ | $60.3 \pm 1.0$ | $81.3 \pm 5.1$ |

Table 6: Robustness of the Gambler's loss to noisy labels, in comparison with the upperbound.

| Dataset | $nll$ loss | Gblers | Gblers$_{AES}$ | Gblers$_{LAES}$ | Upper Bound |
|---|---|---|---|---|---|
| MN $r = 0.2$ | $84.7 \pm 0.5$ | $96.7 \pm 0.2$ | $98.8 \pm 0.1$ (17) | $\mathbf{99.0 \pm 0.1}$ | $98.9 \pm 0.1$ |
| MN $r = 0.5$ | $55.1 \pm 3.1$ | $91.2 \pm 0.7$ | $98.0 \pm 0.0$ (18) | $\mathbf{98.4 \pm 0.2}$ | $98.4 \pm 0.1$ |
| MN $r = 0.65$ | $39.7 \pm 2.5$ | $85.9 \pm 1.1$ | $97.2 \pm 0.2$ (15) | $\mathbf{97.6 \pm 0.3}$ | $98.2 \pm 0.2$ |
| MN $r = 0.8$ | $19.1 \pm 3.0$ | $76.1 \pm 0.3$ | $93.5 \pm 0.3$ (15) | $\mathbf{95.0 \pm 0.5}$ | $98.0 \pm 0.1$ |
| MN $r = 0.85$ | $14.5 \pm 0.7$ | $71.0 \pm 1.4$ | $85.2 \pm 1.7$ (13) | $\mathbf{88.0 \pm 1.0}$ | $97.9 \pm 0.2$ |

corruption is binary), and so we set $o = 2$ in this case. like before, the warmup schedule is for 5 epoch. We see that using Gambler's loss consistently improves the performance at convergence, and using the early stopping criterion gives very strong performance gain over that.

## L    FULL RESULTS ON MNIST

It might be useful to also compare with an experimental upperbound at each corruption rate. See Table 6. The upper bound is computed by only training on $\mathcal{D}_{clean}$ for 100 epochs. We see that while the upper bound is always better than training on the gambler's loss at convergence, performing early stopping on the gambler's loss might even perform better or as good as the upperbound (e.g. when $r = 0.2, 0.5$) because the upper bound, training until the very end, might overfit to the dataset, while early stopping prevents such overfitting.

## M    MORE RESULTS ON IMDB

In this section, we give more results on IMDB, we see that the proposed method outperforms the baseline on all categories we compared with. In this section, $o = 1.9$ for all experiments. In the main body, we have included the results for $r$ in the range $0.1 - 0.3$. Here, we give the results in the range $r = 0.02 - 0.08$. Again, we see that the improvement over $nll$ loss and over VES is consistent, with the gain increasing as the corruption rate increases.

Table 7: Additional Resutls on IMDB dataset.

| Dataset | $nll$ loss | VES | Gblers$_{AES}$ |
|---|---|---|---|
| IMDB $r = 0.02$ | $78.5$ | $79.7$ (25) | $\mathbf{79.8(14)}$ |
| IMDB $r = 0.04$ | $75.4$ | $77.1$ (25) | $\mathbf{78.6\ (13)}$ |
| IMDB $r = 0.06$ | $73.9$ | $74.4$ (20) | $\mathbf{77.6\ (13)}$ |
| IMDB $r = 0.08$ | $73.1$ | $72.9$ (32) | $\mathbf{75.1\ (25)}$ |

