# OpenReview forum: "A Simple Approach to the Noisy Label Problem Through the Gambler's Loss"
_ICLR.cc/2020/Conference — Reject_

### Official Review · AnonReviewer3 · 2019-10-22
**Official Blind Review #3**

**Rating:** 3

**Review:**

This paper proposes a new loss function for dealing with label noise, claiming that the loss function is helpful in preventing overfitting caused by noisy labels. Some experiments show the effectiveness.

The theory part is unclear to me. (1 Why the derivative of Eq. (1) is Eq. (2)? The notation of f, f_\theta, and f_w has been abused. The notation is confusing without explanation. It seems Eq. (2) is not correct and the following is not convincing. (2 Why a small gradient will slower the model to fit the data? This is not clear and maybe not true. (3 The assumptions \hat{p}+\hat{k}+\hat{l}=1 is very strong to me. The events should be dependent. This makes all the theoretical analyses pseudo and not convincing at all. The authors may spend more effort to make the part clear, reasonable, and convincing.

Many claims are very ambiguous. For example, "the key point is that it always holds on some degree", on which degree and why always holds? "In some cases, it even leads to a better memorization phenomenon", in which cases and why lead to better memorization phenomenon?

Some claims are even wrong. For example, "Traditional methods in label noise often involves introducing a surrogate loss function that is specialized for the corrupted dataset at hand and is of no use when one is not aware of the existence of label noise" This is just for some specific methods, not for the most of them.

Typo: "We verify that . Therefore,"

Overall, I cannot understand why the proposed loss function works and cannot recommend acceptance for the current version.


**Experience Assessment:**

I have published in this field for several years.

**Review Assessment: Checking Correctness Of Derivations And Theory:**

I carefully checked the derivations and theory.

**Review Assessment: Checking Correctness Of Experiments:**

I assessed the sensibility of the experiments.

**Review Assessment: Thoroughness In Paper Reading:**

I read the paper thoroughly.

---

> ### Author Response · Authors · 2019-11-07
> **Reply**
>
> Hi! Thank you so much for your comment! We agree that the original writing needs a lot improvement for better clarity, and we have made updates according to your comments. We reorganized and rewrote large parts in section 2, 2.1 and 2.2 to make our theory part clearer. We also corrected sentences that we feel inappropriate.
>
> Now please let me address your questions specifically.
> A1. '3 The assumptions \hat{p}+\hat{k}+\hat{l}=1 is very strong to me. The events should be dependent. This makes all the theoretical analyses pseudo and not convincing at all. The authors may spend more effort to make the part clear, reasonable, and convincing.'
> - We added in section A to explain this. We think it is a misunderstanding. In particular, see equation 12 and the discussion around it. \hat{p}+\hat{k}+\hat{l}=1 is not an assumption and holds by construction. Please also see the current beginning part to section 2 (discussion around equation 1).
>
> A2. '(1 Why the derivative of Eq. (1) is Eq. (2)? The notation of f, f_\theta, and f_w has been abused. The notation is confusing without explanation. It seems Eq. (2) is not correct and the following is not convincing.'
> - Sorry for our sloppy notation in the original version! We have updated this section to make it clearer. Please check the current section 2.2 and let us know if the clarity is improved. We will work hard to make this as clear as possible. Please also see the newly added appendix section B if you are interested in further discussion regarding the robustness of gambler's loss.
>
> A3. (2 Why a small gradient will slower the model to fit the data? This is not clear and maybe not true.
> - Yes, this is indeed an intuitive assumption, and might not hold for some special cases, since zero gradient means no learning has occurred. In fact, please note that the goal of (revised) section 2.2 is to explain, at a high level, a very surprising phenomenon, i.e. training with gambler's loss improves final accuracy when label noise is present. The actual mechanism might be very complicated and task-dependent, and a precise theoretical study of this phenomenon is extremely difficult given our current knowledge about deep learning theory. We added more intuition to this section, and we think that a clear theoretical understanding of this phenomenon is way beyond the scope of this work.
>
> A4. 'Many claims are very ambiguous. For example, "the key point is that it always holds on some degree", on which degree and why always holds?'
> - Sorry for this confusion! We removed this ambiguous sentence and added an example for this. It relates to the fact that on complicated dataset such as CIFAR10, the training accuracy on the clean dataset might be actually significant below 100%.
>
> A5.  'In some cases, it even leads to a better memorization phenomenon", in which cases and why lead to better memorization phenomenon?'
> - There is no such sentence in our paper. In fact, we think this is a misreading of the two neighboring sentences in section 2:
> "As a result, this widens the test accuracy plateau by slowing down the memorization phenomenon. In some cases, it even leads to a better convergence speed and better peak performance (see Figure 4). "
>
> A6. 'Some claims are even wrong. For example, "Traditional methods in label noise often involves introducing a surrogate loss function that is specialized for the corrupted dataset at hand and is of no use when one is not aware of the existence of label noise" This is just for some specific methods, not for the most of them.'
> - Hmmm, thanks for pointing this out. Actually, what we really meant was that 'one common approach amongst others is to introduce a surrogate loss function', since the context was to introduce another loss function approach to the field. We have corrected this sentence.
>
> A7. We have also fixed the typo you mentioned.
>
> Again, thank you so much for the review, and please check the newly revised section 2 and section A B in the appendix for your questions. Please let us know if you have more questions regarding section 2, and we will do our best to make it as clear as possible.

---

> ### Author Response · Authors · 2019-11-13
> **Experiment Updates**
>
> Hi! We just updated the experiment sections as well. The updated experiments were suggested by the other reviewers and we think helped with making the paper more solid and demonstrated the effectiveness of the proposed method. Please let me summarize the update to the experiments as follows:
>
> (1) we updated Table 1 in section 4.2 to study our method on the IMDB dataset, which is a standard NLP task for sentiment analysis. The model used is a standard LSTM. We have updated the experiment section of our paper to include experiment on the IMDB dataset, see table 1 (r=0.1-0.3) and also see appendix section M (r=0.02-0.08), where the gambler’s loss is shown to outperform the baseline and the AES criterion is shown to outperform validation early stopping method significantly.
>
> (2) section J: we include two more demonstrations of the three stage phenomenon. One is on IMDB, using LSTM and GloVe word embedding, and the other is on openimage which is a noisy dataset whose noise rate is hard to estimate), for this example, we also show how the proposed criterion might help to estimate the noise level in the dataset
>
> (3) section K: we include some experiment to demonstrate that the proposed method might also be robust to asymmetric noise; the improvement over the baseline is consistent and significant; however, given the time and computational resource we had, the experiments are small scale; moreover, we actually plan to deal with the asymmetric label noise in a future work
>
> (4) section L: we compare our method to the “upper bound”, i.e. training without the corrupted data (no early stopping). We notice that for some range (r=0.2-0.5), analytical early stopping perform at least as good as the upper bound, we hypothesize that this is because our method also effective prevents overfiting by stopping early
>
> (5) Besides, we removed the part that combine our method with CT (co-gambling), because, as argued in the original version, the two methods do not seem very compatible. In its place, we added a simple schedule to improve our method consistently, called LAES, which starts the training by five epochs of warmup (o=m) and then switch to smaller o.
>
> (6) we also updated the CIFAR10 experiments in table 2 to include a comparison with simple training the nll loss, and we thinks this further shows the effectiveness of our method. While the proposed method does not achieve SOTA results in the range (r=0.2-0.6), it still achieves significant improvement over simply training on the nll loss (by 10-30% absolute accuracy, see the updated table 2), and we think this deserves some merit. In comparison with methods such as CT, we note that CT outperforms the proposed method only marginally by about 2-5% accuracy in the rage (r=0.2-0.6) while requiring twice as many parameters and training time, and so we think another merit of the proposed method is its simplicity.
>
>
> While the theory part bases on assumptions that seem quite strong, we think the above experiments further verified these assumptions, and the fact that the proposed method is shown to be effective on these datasets and tasks further suggest the correctness and wide applicability of the theory.

---

### Official Review · AnonReviewer1 · 2019-10-22
**Official Blind Review #1**

**Rating:** 3

**Review:**

Updated review: Thanks for your comments. I feel the latest version of the paper is better than the previous version.
However, as stated by other reviewers as well, the claims of the paper are quite ambiguous. Another example from the author response is the point about how Chapter 6 of the Elements of Infomation Theory is related to Gambler's loss. This is not clear to me. I would not object to accepting the paper but I find it difficult to recommend accept for this paper. Perhaps the authors can be more clear in their claims.

----------------------------------------------------------------------------------------------------------------------------------------------------------

Summary: The paper focusses on the problem of noisy labels in supervised learning with deep neural networks. The paper, in turn, proposes an early stopping criterion for handling label noise. The early stopping criterion is dependent on a new loss function that is defined as the log of true label + weight on a reservation option? The paper shows that when the labels are corrupted then the propose early stopping criterion does better than early stopping criteria obtained via the validation set.

\The first section of the paper establishes that when label noise is present in the dataset, then there are three stages to training a deep neural network.
The learning stage where the highest accuracy on the test set is achieved.
The gap stage where test set accuracy goes down.
Memorization stage corresponds to when a deep neural network memorizes corrupt labels and test accuracy goes completely down.
I cannot understand figure 1(a). The y-label says accuracy but it seems that the plot is about loss. What dataset was this and what architecture of DNN was used? The plot shows that the DNN achieved a 100% accuracy in 5 epochs. Is this result meaningful? Before establishing a hypothesis based on this should the hypothesis not be tested on multiple datasets.
The paper says that these stages are persistent across multiple architectures and datasets and as proof the paper says ‘we verified that’. Why can’t the reader see the experiments? By across datasets does the paper mean MNIST and CIFAR? By across architecture does the paper mean the two architectures mentioned in the appendix one each for MNIST and CIFAR respectively?

The paper makes the assumption that label noise is symmetrically corrupted. Why and where does such an assumption hold? What happens to the proposed method if that is not true.

Assumption 2: During the gap stage the model has learned nothing about the corrupt data points.
How is that even possible?

Equation 1: So the loss function proposed is log(f(x)_y + (1/o) f(x)_m+1) .  What is y here? The true label? Why is y called a point mass? Is this different from the cross-entropy loss + log loss on m+1 ?

I do not understand equations 2 to 5.

For figure 3 again what datasets were used?

“ Making random bet will help with making money and a skilled gambler will not make such bets”
Why does making random bet help with making money? If random is good how can a skilled gambler exist in such a game? What is this skill?

k denotes the sum of probability of predicting anything that is not y or m+1 (it does not denote prediction).

In the experiments section what was the symbol for the rate of corruption changed from epsilon to r. Are they different?

What is nll?

It seems that gamblers loss best shines when the corruption rate is as high as 80% . That is 80 percent of the data is corrupted. Does this mean that if I trained with only 20% of the non-corrupt data I would still get a 99% accuracy on MNIST (even without gamblers loss)? A comparison of this sort would have been useful.
One astonishing result the paper presents is that with gambler’s loss even with 80% corrupt labels a 94% test accuracy is possible on MNIST dataset. I think this is significant, this raises the question that is it required to label all the data points ina dataset to achieve high accuracy or is it possible to achieve just as much with only 20% of the labels?

**Experience Assessment:**

I have read many papers in this area.

**Review Assessment: Checking Correctness Of Derivations And Theory:**

I assessed the sensibility of the derivations and theory.

**Review Assessment: Checking Correctness Of Experiments:**

I assessed the sensibility of the experiments.

**Review Assessment: Thoroughness In Paper Reading:**

I read the paper at least twice and used my best judgement in assessing the paper.

---

> ### Author Response · Authors · 2019-11-08
> **Reply (part 1)**
>
> Hi! Thank you so much for the comment! I have read your comment carefully and I think, fortunately, many questions are due to misunderstanding, and we have updated the manuscript to make many points more clear (this includes a rewriting of section 2, and addition of section A and section B in the appendix). Before adding in experiments or rewriting sections, let me answer the questions that we can already provide answer to. We will let you know once we update the experiment section.
>
> A1. 'I cannot understand figure 1(a). The y-label says accuracy but it seems that the plot is about loss.'
> -For figure 1a, please note that figure 1a has two y-axes; on the left it says accuracy, and on the right it says loss, also, please note that we consistently we used this 2-axes style in the paper (e.g. Fig.3, Fig.5, Fig.6).
>
> A2. 'What dataset was this and what architecture of DNN was used?'
> -As is described in the title of Figure 1. This plot is for MNIST with corruption rate 0.5. Since the dataset is MNIST, the architecture is the one described in Table 3 in the appendix.
>
> A3. 'The plot shows that the DNN achieved a 100% accuracy in 5 epochs. Is this result meaningful?'
> -Hmmm, it is a little hard to answer this question without a clear definition of 'meaningful'. The dataset is MNIST, and usually the accuracy reaches >95% using a DNN without label noise.
>
> A4. 'Before establishing a hypothesis based on this should the hypothesis not be tested on multiple datasets.The paper says that these stages are persistent across multiple architectures and datasets and as proof the paper says ‘we verified that’. Why can’t the reader see the experiments? By across datasets does the paper mean MNIST and CIFAR? By across architecture does the paper mean the two architectures mentioned in the appendix one each for MNIST and CIFAR respectively? '
> -In addition to the two CNN architectures we described in the paper, two experiments are done using ResNet18 (please see Figure 5.c and 5.d, and search for the word ResNet). In fact, this 3-stage phenomenon is quite easily observable for many datasets when decent level of label noise is present. Since the paper is already 16 pages long, we did not include more experiments in the initial version, but as you suggested, we will also include a few plots from other dataset and architectures soon (we will let you know once we update the experiments).
> -- 'we verified that .' was actually a type, we have removed this sentence.
>
> A5. 'The paper makes the assumption that label noise is symmetrically corrupted. Why and where does such an assumption hold? What happens to the proposed method if that is not true. '
> - Since the criterion does not apply to assymetrically corrupted data, to deal with asymmetrically corrupted data, some other criterion is needed. However, simply using gambler's loss in this case should also improve the final performance. We can also update some toy experiment on this if time allows by the time the rebuttal period ends.
>
> A6. 'Assumption 2: During the gap stage the model has learned nothing about the corrupt data points. How is that even possible? '
> - Empirically, the situation is that, on average, the model makes random prediction on the corrupted dataset. So there might be some corrupt points that the model actual learned, but the number of such points should be small compared to the ones the model has not learned. Moreover, this assumption can be verified by Figure 1, where the training loss on the corrupt part does not start to decrease significantly until 20th epoch, where learning of the clean data is very low. Theoretically speaking, this is because gradient descent tends to learn function of increasing complexity, and the corrupt labels constitute a very high complexity function, and so is very hard to learn compared with clean points [2].

---

> > ### Author Response · Authors · 2019-11-08
> > **Reply (part 2)**
> >
> > A7. 'Equation 1: So the loss function proposed is log(f(x)_y + (1/o) f(x)_m+1) .  What is y here? The true label? Why is y called a point mass? Is this different from the cross-entropy loss + log loss on m+1 ?'
> > - Sorry for making this confusing! We have rewritten section 2 to make this as clear as we can. y is indeed the true label. Saying that y is a point mass simply means that it is a 0-1 loss (having value 1 for the correct label, and having 0 for the incorrect labels). The loss function function indeed takes the form log(f(x)_y + (1/o) f(x)_m+1), this has well studied information-theoretic properties (see chapter 6 of [1] for a detailed discussion on its intuitive meaning and mathematical properties). The function you mentioned might work in practice? But it is hard to give interpretation on such loss, and its theoretical properties are, to our knowledge, yet unknown.
> >
> > A8. 'For figure 3 again what datasets were used?'
> > -As we mentioned in the paper, the exact detail for figure 3 is given in the appendix section I, and as is described there, it is done in MNIST with corruption rate 0.5. Many more experiments are also shown there.
> >
> > A9. 'I do not understand equations 2 to 5. '
> > - Reviewer 3 also found this part a little confusing, and we agree that our original presentation of this part can indeed be greatly improved. We have reorganized and rewrote section 2 and added some intuitions of the mechanism at working (see the short paragraph above the current equation 5), in the hope to make this clearer. We also added appendix section A and appendix section B to clalrify further details. Please let us know if you still have any question about this part.
> >
> > A10. '“ Making random bet will help with making money and a skilled gambler will not make such bets”Why does making random bet help with making money? If random is good how can a skilled gambler exist in such a game? What is this skill?'
> > - Sorry! This is a typo! It should be “ Making random bet will NOT help with making money and a skilled gambler will not make such bets"
> >
> > A11. 'k denotes the sum of probability of predicting anything that is not y or m+1 (it does not denote prediction). '
> > - Hmmm, what we really meant was that 'k denotes the sum of PREDICTED probability of predicting anything that is not y or m+1'. We have updated section 2.1 to clarify this. Please also see appendix section A for more detailed information about the gambler's loss.
> >
> > A12. 'In the experiments section what was the symbol for the rate of corruption changed from epsilon to r. Are they different? '
> > - epsilon refers to (1-r), meaning the non-corrupt rate, so they are two different symbols.
> >
> > A13. 'What is nll? '
> > - nll is shorthand for negative log loss, which is also called cross entropy loss
> >
> > Please let us know if you find any other parts confusing! We will do our best to revise the manuscript and clarify the points.
> >
> > [1] Thomas M. Cover. Elements of Information Theory (Wiley Series in Telecommunications and Signal Processing).
> > [2] https://arxiv.org/abs/1905.11604

---

> ### Author Response · Authors · 2019-11-13
> **Experiment Updates**
>
> Hi! We have updated our experiment section to answer your questions! Indeed, we think that adding in these experiments make the current paper more solid. The update to experiments include:
>
> (1) section J: we include two more demonstrations of the three stage phenomenon. One is on IMDB, using LSTM and GloVe word embedding, and the other is on openimage which is a noisy dataset whose noise rate is hard to estimate), for this example, we also show how the proposed criterion might help to estimate the noise level in the dataset
>
> (2) section K: we include some experiment to demonstrate that the proposed method might also be robust to asymmetric noise; the improvement over the baseline is consistent and significant; however, given the time and computational resource we had, the experiments are small scale; moreover, we actually plan to deal with the asymmetric label noise in a future work
>
> (3) section L: we compare our method to the “upper bound”, i.e. training without the corrupted data (no early stopping). We notice that for some range (r=0.2-0.5), analytical early stopping perform at least as good as the upper bound, we hypothesize that this is because our method also effective prevents overfiting by stopping early
>
> (4) moreover, we also updated Table 1 in section 4.2 to study our method on the IMDB dataset, which is a standard NLP task for sentiment analysis. The model used is a standard LSTM. We have updated the experiment section of our paper to include experiment on the IMDB dataset, see table 1 (r=0.1-0.3) and also see appendix section M (r=0.02-0.08), where the gambler’s loss is shown to outperform the baseline and the AES criterion is shown to outperform validation early stopping method significantly.
>
> (5) Besides, we removed the part that combine our method with CT (co-gambling), because, as argued in the original version, the two methods do not seem very compatible. In its place, we added a simple schedule to improve our method consistently, called LAES, which starts the training by five epochs of warmup (o=m) and then switch to smaller o.
>
> Let us also provide an answer the the following question:
> A. 'It seems that gamblers loss best shines when the corruption rate is as high as 80% . That is 80 percent of the data is corrupted. Does this mean that if I trained with only 20% of the non-corrupt data I would still get a 99% accuracy on MNIST (even without gamblers loss)? A comparison of this sort would have been useful.  '
>
> - we indeed think that the gambler's loss can be very useful when the strength of noise present is very strong. Since the current methods can hardly deal with extremely strong label noise (>0.7) especially because the noise rate becomes very hard to estimate at this stage. However, we also argue that the value for the proposed method should not be underestimated when the noise rate is small. For example, on CIFAR10, while the proposed method does not achieve SOTA results in the range (r=0.2-0.6), it still achieves significant improvement over simply training on the nll loss (by 10-30% absolute accuracy, see the updated table 2), and we think this deserves some merit. In comparison with methods such as CT, we note that CT outperforms the proposed method only marginally by about 2-5% accuracy in the rage (r=0.2-0.6) while requiring twice as many parameters and training time, and so we think another merit of the proposed method is its simplicity.

---

### Official Review · AnonReviewer2 · 2019-10-23
**Official Blind Review #2**

**Rating:** 6

**Review:**


Update after rebuttal:

The good:
The rebuttal and updated paper address many of my concerns. Most importantly, the updated paper demonstrates the three-stage phenomenon on Open Images and adds experiments on IMDB showing that the Gambler's loss with AES helps a lot. The LAES iteration introduced in the updated paper alleviates my concern about performance drop compared to the CT baseline at certain corruptions on CIFAR-10.

The bad:
- From Figure 12, it looks like the three-stage phenomenon doesn't hold on IMDB. Does AES provide additional benefit beyond the Gambler's loss on IMDB? This needs to be clarified with the way Figure 12 turned out.
- There is a serious missing citation [1] that should be included as a baseline. The proposed method in [1] is at least superficially similar to the Gambler's loss and also makes use of the fact that it is easier to fit clean labels than noisy labels. My apologies for not noticing this earlier.

Overall:
I would still suggest acceptance, because the three-stage phenomenon is an interesting find that the authors make good use of. In light of the missing citation, though, I cannot raise my score.


[1]: Zhilu Zhang, Mert R. Sabuncu. "Generalized Cross Entropy Loss for Training Deep Neural Networks with Noisy Labels". NeurIPS 2018.

-----------------------------------------------------------------------
Summary:
This paper proposes a method to alleviate label noise. It opens with the observation of three distinct stages when training in the presence of label noise. Importantly, there is a ‘gap’ stage during which the network has not begun memorizing noisy labels and early stopping is ideal. The authors then observe that the Gambler’s loss (Ziyin et al., 2019) elongates the gap stage and propose an analytic early stopping (AES) criterion for identifying when to stop training.

The analysis of the AES criterion, e.g. in Figure 5, and the observation of a phase transition when tuning the o hyperparameter are quite interesting, and the latter observation is of practical value when using the AES criterion.

The AES criterion seems to be well-motivated, and the empirical evaluation of the Gambler’s loss with and without early stopping is good. The results are strong on MNIST but somewhat weak on CIFAR-10. Specifically, the improvements on CIFAR-10 only appear for large corruption rates (0.7+), and performance is lower than the baselines for other corruption rates. This is a worrying problem, because it calls into question the value of the method on larger problems. However, seeing as this is a distinct approach from the baselines and that it demonstrates some promise, I recommend borderline accept. The authors could raise my score by demonstrating more consistent gains on another larger-than-MNIST CV dataset or an NLP/speech dataset. Other points of concern that I have are listed below.

Major points:
At the top of page 3, the authors say that the idealized gap assumption “holds well for simple datasets such as MNIST and on datasets with very high corruption rate, where our method achieves best results, and less so on more complicated datasets such as CIFAR10”. The idealized gap assumption is behind the AES criterion, but Figure 5 suggests that the AES criterion works well on CIFAR-10, so what do the authors mean when they say the assumption doesn’t work as well on CIFAR-10? Is this just referring to the results?

Saying traditional label noise correction methods are “of no use when one is not aware of the existence of label noise” seems unfair. The FC method and others do not require foreknowledge of the corruption rate and do not harm performance in the absence of label noise, so they can also be said to automatically correct label noise.

“FC, however, requires knowing the whole transition matrix, and is outperformed significantly by our method.”
This is not quite true, because Patrini et al. propose an estimate of the transition matrix as part of the Forward correction. Did you use the estimated or true transition matrix for the FC method? It would be good to clarify this in the paper.

Minor points:
There are a few grammatical errors and typos in the paper:

“or explicit regularization, this is also what is suggested by Abiodun et al. (2018)” (run-on sentence)\

“CIFAR10” should be “CIFAR-10”

**Experience Assessment:**

I have published one or two papers in this area.

**Review Assessment: Checking Correctness Of Derivations And Theory:**

I did not assess the derivations or theory.

**Review Assessment: Checking Correctness Of Experiments:**

I carefully checked the experiments.

**Review Assessment: Thoroughness In Paper Reading:**

I read the paper at least twice and used my best judgement in assessing the paper.

---

> ### Author Response · Authors · 2019-11-13
> **Reply and experiment updates**
>
> Hi! Thank you so much for your reply! I think your advice really helped us improving the paper. We have updated the experiment section of our paper to include experiment on the IMDB dataset, see table 1 (r=0.1-0.3) and also see appendix section M (r=0.02-0.08), where the gambler’s loss is shown to outperform the baseline and the AES criterion is shown to outperform validation early stopping method significantly. Besides, we removed the part that combine our method with CT (co-gambling), because, as argued in the original version, the two methods do not seem very compatible. In its place, we added a simple schedule to improve our method consistently, called LAES, which starts the training by five epochs of warmup (o=m) and then switch to smaller o.
>
> Besides this part, other updates to the experiments include: (1) section J: we include two more demonstrations of the three stage phenomenon (on IMDB and on openimage); (2) section K: we include some experiment to demonstrate that the proposed method might also be robust to asymmetric noise; (3) section L: on MNIST, we compare our method to the “upper bound”, i.e. training without the corrupted data. If you are interested , we also included two sections to elaborate on our theory part (Section A and section B).
>
> Now please let me address your specific questions.
>
> A1. ‘Specifically, the improvements on CIFAR-10 only appear for large corruption rates (0.7+), and performance is lower than the baselines for other corruption rates. This is a worrying problem, because it calls into question the value of the method on larger problems.’
> - While the proposed method does not achieve SOTA results in the range (r=0.2-0.6), it still achieves significant improvement over simply training on the nll loss (by 10-30% absolute accuracy), and we think this deserves some merit. In comparison with methods such as CT, we note that CT outperforms the proposed method only by 2-5% accuracy in the rage (r=0.2-0.6) while requiring twice as many parameters and training time, and so we think another merit of the proposed method is its simplicity.
>
> A2. ‘At the top of page 3, the authors say that the idealized gap assumption “holds well for simple datasets such as MNIST and on datasets with very high corruption rate, where our method achieves best results, and less so on more complicated datasets such as CIFAR10”. The idealized gap assumption is behind the AES criterion, but Figure 5 suggests that the AES criterion works well on CIFAR-10, so what do the authors mean when they say the assumption doesn’t work as well on CIFAR-10? Is this just referring to the results?’
> - Sorry for this confusion! The same problem was also pointed to by reviewer 3. We removed this ambiguous sentence and added an example for this. It relates to the fact that on complicated dataset such as CIFAR10, the training accuracy on the clean dataset might be actually significant below 100%. While for MNIST, the training accuracy on the clean part is very close to 100%.
>
> A3. ‘Saying traditional label noise correction methods are “of no use when one is not aware of the existence of label noise” seems unfair. The FC method and others do not require foreknowledge of the corruption rate and do not harm performance in the absence of label noise, so they can also be said to automatically correct label noise.’
> - Yes, it is true that other methods, such as FC, when used without label noise, can be seen as a simple label smoothing method and should not harm learning. What we really meant was the cases in which the noise rate might be small (say, r=0.02-0.08) and might be hard to estimate, and in these case, other methods are unlikely to provide improvement when r is unknown. However, using the gambler’s loss with some benign value of o is observed to actually improve the performance of the model (compared to training with nll loss). For example see section M (where o is set to 1.9 without any tuning).
>
> A4. ‘“FC, however, requires knowing the whole transition matrix, and is outperformed significantly by our method.”This is not quite true, because Patrini et al. propose an estimate of the transition matrix as part of the Forward correction. Did you use the estimated or true transition matrix for the FC method? It would be good to clarify this in the paper.’
> - Yes, we used the estimation method proposed in the original paper, as we described in the related works section. When the transition matrix is exactly known, the Forward correction method can actually be quite strong (or even the best method).
>
> A5. We corrected the minor points you mentioned.
>
> Again, thank you very much for the comments, and please let us know if you have any other questions!

---

### Public Comment · ~Yilun_Xu1 · 2019-11-04
**Related work/baseline missing**

Hi,

[1] proposes the first loss function that is provably not sensitive to noise patterns and noise amount. Also, [1] does not require to know the noise patterns or noise amount beforehand.  I wonder how the noise-robust function[1] performs in your setting. In experiments of [1], the noise pattern has both symmetry and asymmetry patterns, as well as diagonal-dominant and non-diagonal-dominant patterns.

Thank you!


[1] Yilun Xu, Peng Cao, Yuqing Kong, and Yizhou Wang. L_DMI: A novel information-theoretic loss function for training deep nets robust to label noise.  NeurIPS 2019

---

> ### Author Response · Authors · 2019-11-06
> **Reply**
>
> Hi Yilun! Thanks for your comment and for noticing our work! The paper you pointed us to is indeed interesting, and using determinant of the estimated joint matrix is very novel and seems to work very well (I personally learned a lot from reading your paper). We were not aware of this paper before and we think it is good for us to relate to this work in our paper. However, we decided that we will refrain from making a comparison with this method for the following two reasons: (1) let C denote the number of classes, then computing the determinant of the joint matrix is of O(C^3) complexity (e.g., for the numpy implementation), for tasks such as CIFAR100 or ImageNet, it does not look like the method will scale up easily, this is then followed by a matrix inversion, which is again of Omega(C^2.3) complexity; (2) the loss function also seems to have non-trivial batchsize dependence in order to make the estimated joint matrix full rank, for Imagenet, for example, this method needs at least 1000 batchsize, and to ensure that the matrix is full rank with high probability, the batchsize seems to need to be another order of magnitude larger. In short, we really want to compare with methods with similar computational complexity, and we will relate to this work in the paper. Still, we are very excited to hear about it.

---

### Decision · Program_Chairs · 2019-12-19

**Decision:**

Reject

**Comment:**

This paper focuses on mitigating the effect of label noise. They provide a new class of loss functions along with a new stopping criteria for this problem. The authors claim that these new losses improves the test accuracy in the presence of label corruption and helps avoid memorization. The reviewers raised concerns about (1) lack of proper comparison with many baselines (2) subpar literature review and (3) state that parts of the paper is vague. The authors partially addressed these concerns and have significantly updated the paper including comparison with some of the baselines. However, the reviewers were not fully satisfied with the new updates. I mostly agree with the reviewers. I think the paper has potential but requires a bit more work to be ready for publication and can not recommend acceptance at this time. I have to say that the authors really put a lot of effort in their response and significantly improved their submission during the discussion period. I recommend the authors follow the reviewers' suggestions to further improve the paper (e.g. comparing with other baselines) for future submissions